# The Role of Ticks in the Emergence of *Borrelia burgdorferi* as a Zoonotic Pathogen and Its Vector Control: A Global Systemic Review

**DOI:** 10.3390/microorganisms9122412

**Published:** 2021-11-23

**Authors:** Sabir Hussain, Abrar Hussain, Umair Aziz, Baolin Song, Jehan Zeb, David George, Jun Li, Olivier Sparagano

**Affiliations:** 1Department of Infectious Diseases and Public Health, Jockey Club College of Veterinary Medicine and Life Sciences, City University of Hong Kong, Kowloon, Hong Kong, China; umair.uvas@gmail.com (U.A.); Baolin.Song@my.cityu.edu.hk (B.S.); zebjehan2012@gmail.com (J.Z.); jun.li@cityu.edu.hk (J.L.); 2Department of Epidemiology and Public Health, University of Veterinary and Animal Sciences, Lahore 54600, Pakistan; Abrar.arid@gmail.com; 3School of Natural and Environmental Sciences, Newcastle University, Newcastle upon Tyne NE1 7RU, UK; david.george1@newcastle.ac.uk

**Keywords:** ticks, tick-borne disease, *Borrelia burgdorferi*, Lyme disease, zoonoses, control

## Abstract

Ticks are widely distributed across the globe, serving as hosts for numerous pathogens that make them major contributors to zoonotic parasitosis. *Borrelia burgdorferi* is a bacterial species that causes an emerging zoonotic tick-borne disease known as Lyme borreliosis. The role of ticks in the transmission of this pathogen was explored in this study. According to this systematic review, undertaken according to Preferred Reporting Items for Systematic Reviews and Meta-Analyses (PRISMA) guidelines, 19 tick species are known to carry *Borrelia burgdorferi*, with more than half of the recorded cases in the last two decades related to *Ixodes ricinus* and *Ixodes scapularis* ticks. Forty-six studies from four continents, Europe, North America, Asia, and Africa, reported this pathogen in ticks collected from vegetation, animals, and humans. This study highlights an increasing distribution of tick-associated *Borrelia burgdorferi*, likely driven by accelerated tick population increases in response to climate change coupled with tick dispersal via migratory birds. This updated catalogue helps in compiling all tick species responsible for the transmission of *B. burgdorferi* across the globe. Gaps in research exist on *Borrelia burgdorferi* in continents such as Asia and Africa, and in considering environmentally friendly vector control strategies in Europe and North America.

## 1. Introduction

Globally, one-third of the emergence in infectious diseases during last two decades is due to zoonotic vector-borne diseases, which have major devastating effects on human and veterinary health and welfare [1]. Especially, the incidence of Lyme disease in USA is expected to increase by about 20% in the next 1 or 2 decades due to climate change [2]. Ticks are considered as the second-most threatening vector for human health after mosquitoes, transmitting various pathogens [3]. Transmission channels of tick-borne infections must be well understood to mitigate livestock production losses and impacts on animal welfare and reduce disease exposure in humans [4]. This is especially important given that tick-borne zoonosis is increasing in the twenty-first century, driven largely by climate change impacts on tick lifecycles and the transboundary movement of tick-infested animals [5].

Lyme borreliosis (LB) is an example of a significant, and increasing, tick borne zoonosis, caused by the *Borrelia burgdorferi* sensu lato *(s.l.)* complex [6]. New species are still being identified within this complex, which contains 21 species to date [7]. It is primarily comprised of *Borrelia burgdorferi* sensu stricto *(s.s.), Borrelia afzelii*, and *Borrelia garinii*, which are common in Europe and Asia, and *B. burgdorferi* in North America but is also associated with other unknown pathogens that pose a threat to human health [8,9]. The overall burden of *Borrelia burgdorferi (B. burgdorferi)* is poorly understood [10], despite the CDC reporting over 476,000 cases annually in the United States alone [11]. A study conducted in Europe estimated more than 200,000 *B. burgdorferi* infections in humans annually [12].

*Borrelia burgdorferi* infection occurs in a wide variety of animals, along with humans, including small wild mammals and birds [13]. Typically, uninfected six-legged larvae feed on infected small mammals (domestic and wild) or birds, moulting into an infected eight-legged nymph, though unfed larvae are also a source of transovarial transmission. During the period of transformation from nymph to adult, infested ticks will feed upon small mammals, domestic animals, or humans as secondary hosts, potentially resulting in *B. burgdorferi* transmission. Adult ticks then seek a final host for mating, which may be either white-tailed deer (in case of *Ixodes scapularis*) or other animals, including humans [14]. If infected, *Borrelia burgdorferi* in humans causes fatigue, fever, musculoskeletal pain, erythema migrans, and the potential for cardiac and neurological symptoms, with an incubation period of 3–30 days [15]. In the case of Europe, Lyme neuroborreliosis (10–15% cases) is the neurological sign that may be observed in early stages of Lyme disease [16]. The most common sign observed is Bannwarth syndrome in Europe, in which individuals feel intense nerve pain radiating from the spine. That situation is uncommon in North America [9,17], though; arthritis is the most common complication observed in the US which is rarely seen in Europe [9].

*Borrelia burgdorferi* is the most prevalent tick-borne pathogen in temperate regions of the Northern Hemisphere, but the expansion of geographical boundaries of ticks makes this pathogen a significant health concern worldwide [6]. Ticks as a vector play a central role in geographical disease expansion and host-to-host transmission of *B. burgdorferi* [18]. This pathogen is vectored by the genus *Ixodes*, commonly by *Ixodes scapularis* (*I. scapularis, Ixodes ricinus, Ixodes persulcatus*, and *Ixodes pacificus*) species, but with other members of this genus also contributing to transmission [19]. The species of vector determines the range of host availability for *B. burgdorferi*, which can significantly affect transmission dynamics [20].

At present, attempts to control the transmission of Lyme disease relies on targeting ticks directly. Many of these control strategies give cause for concern, however, they risk damage to the natural environment through widespread deployment of various acaricides [21], and often only target ticks during an isolated stage of their lifecycle. Effective control of tick-borne pathogens can only be achieved if delivered to consider interrelated human, animal, and ecological perspectives, but the deployment of holistic approaches is hard to implement. For instance, biodiversity protection and creation of urban green areas are crucial for animal and human health but increase the burden of tick-borne diseases (TBDs). In contrast, a decreased population of wild animals carrying ticks, or implementation of measures such as fencing to exclude them from certain areas, will reduce the transmission of TBDs, but could have devastating effects on biodiversity that are socially unacceptable [22].

In this study, we aimed to systematically analyze the research on *B. burgdorferi* in ticks during the first two decades of the twenty-first century. We focused on the prevalence rate of ticks carrying *B. burgdorferi*, the method of detection, location of cases, and the changes in prevalence over time. In this way, we highlight the emerging trend of this zoonotic agent through ticks worldwide, also suggesting preventive strategies for its control.

## 2. Materials and Methods

### 2.1. Study Protocol

We collected data following Preferred Reporting Items for Systematic Reviews and Meta-Analyses (PRISMA) guidelines [23]. Using this format, we systemically reviewed the relevant literature on *Borrelia burgdorferi* in ticks across the world.

### 2.2. Data Sources and Search Strategy

In the data-retrieving process, five search engines, namely Scopus, PubMed, Google Scholar, Science Direct, and Web of Science, were used between 1 March 2021 and 8 August 2021 to search for articles published on *Borrelia burgdorferi* in ticks from 1 January 2001 to 15 June 2021. The keywords used for the search included tick(s), zoonosis, borreliosis, and *Borrelia burgdorferi*. We used the library online search database of the City University of Hong Kong. We requested access to those articles also which were not available online with full text.

### 2.3. Data Extraction

To maximize accuracy, information was extracted and compiled in Microsoft Excel 2019 by two authors (S.H. and A.H.) independently, screened to remove repeated studies in individuals’ files, and then merged to avoid duplication. The discrepancy in extracted data from both authors (S.H. and A.H.) was double-checked by the third author (O.S.) and discussed to create the relevant article lists, which included authors, study title, year of publication, journal name, volume, issue, page number, DOI, author affiliations, abstract, and keywords.

### 2.4. Study Selection Criteria

Our screening strategy consisted of two steps. First, titles, abstracts, and keywords were used to eliminate duplicates, extraneous review studies, and those not published in English. In the second step, the full text of all relevant studies was thoroughly reviewed to screen and extract the necessary details. The key features that were taken into account for the inclusion of studies in the second step were (i) study included the detection of *Borrelia burgdorferi* in ticks (with tick species detection) or tick-infested animals/humans, (ii) study provided details about positive cases and total samples, (iii) study stated the location of sample collection sites, and (iv) study mentioned the techniques used for detection.

### 2.5. Quality Assessment and Selection

During the first step of screening, data compiled in Microsoft Excel 2019 files by two authors independently included 574 and 603 studies, resulting in a total of 734 studies after merging into a single file. The third author removed the duplicate studies (*n* = 25). Subsequent screening on the basis of titles, abstracts, and keywords removed a further 390 studies, which were not focused on *Borrelia burgdorferi*, but on other aspects of Lyme disease, followed by removal of another 47 that did not contain original research (e.g., review articles, meta-analysis, and opinion pieces) to avoid the repetition in reported data. Only one study was removed in a language other than English (Chinese), with 152 studies excluded because they did not investigate ticks or tick-infested animals/humans for pathogen detection; rather, they focused on seroprevalence of *Borrelia burgdorferi* without including any history related to ticks. Those studies (*n* = 52) which did not investigate the prevalence of pathogen, and instead investigated some ecological, biological, and evolutionary aspects of the pathogen, were also eliminated, as were three studies that did not mention an exact location for sampling, and 18 studies where diagnostic methods were not given. A total of 46 studies were finalized, their references were reviewed by authors, and data from those studies were arranged in tabular form using Microsoft Word 2019, with details included for the title of study, year of data collection, year of publication, sites for sample collection, country of study, the continent of study, number of positive samples, total samples, prevalence, confidence interval (CI 95%), technique used, and reference of study (Figure 1).

## 3. Results and Discussion

### 3.1. Spatial Distribution of Borrelia burgdorferi and Ticks

Several species of ticks reported positive for *Borrelia burgdorferi* from studies conducted in the previous two decades. The majority of these studies reported positive cases in *Ixodes ricinus* (*n* = 33; 71.7%) and *Ixodes scapularis* (*n* = 9; 19.5%), followed by the other 17 tick species shown in Figure 2. In many studies, more than one tick species was found to be positive for this pathogen; therefore, the percentages (Figure 2) are given with this in mind. The spatial distribution of *B. burgdorferi* in ticks was found to be global in nature, with *B. burgdorferi* reported from ticks in four out of the seven continents, namely Europe, North America, Asia, and Africa. The highest proportion of studies reported cases from Europe (*n* = 34; 73.9%) followed by North America *(n* = 9; 19.6%), Asia (*n* = 2; 4.3%), and Africa (*n* = 1; 2.2%). At a national level, the highest proportion of studies reporting *Borrelia burgdorferi* in ticks were from Italy (*n* = 8; 17.4%) and the US 17.4% (*n* = 8; 17.4%) (Figure 3).

### 3.2. Distribution/Prevalence of Borrelia burgdorferi in Ticks of Different Continents

#### 3.2.1. Europe

In the previous two decades, 34 studies have been conducted that report *Borrelia burgdorferi* from European ticks. Of these, the highest number of studies were from Italy (*n* = 8) followed by Finland (*n* = 3), the Netherlands (*n* = 2), Slovakia (*n* = 2), Serbia (*n* = 2), Romania (*n* = 2), Ukraine (*n* = 2), Belarus (*n* = 2), Latvia (*n* = 2), Sweden (*n* = 1), Germany (*n* = 1), Scotland (*n* = 1), England (*n* = 1), England and Scotland (*n* = 1), Denmark (*n* = 1), Luxembourg (*n* = 1), Switzerland (*n* = 1), Poland (*n* = 1), and Czech Republic, Estonia, Germany, Greece, Hungary, Netherland, Portugal, Slovenia, Spain, and Sweden (all combined) (*n* = 1). Out of these 34, in 44.1% (*n* = 15) of studies, ticks were collected from vegetation (e.g., parks, forest, and hilly areas), while 38.2% (*n* = 13) of studies involved collection of ticks from wild and domestic animals, and 17.6% (*n* = 6) collected ticks from both vegetation and animals (Figure 4 and Figure 5; Table 1). Almost 97% (*n* = 33) of studies conducted in Europe identified *Ixodes ricinus* ticks as being positive for *Borrelia burgdorferi*, which indicates that this species is the major transmitting source of this pathogen in this region. PCR was used to confirm the presence of *Borrelia burgdorferi* in all 34 studies, and a 17.7% prevalence of *Borrelia burgdorferi* was found in European ticks when taking the average of the prevalence mentioned in all studies.

The consideration of associations of *B. burgdorferi* with ticks and the environment is inevitable. Thus, any change in these associated factors will create a major impact on this pathogen’s distribution, and unexpected consequences may result [70]. According to our literature review, the tick species most associated with *B. burgdorferi* was *I. ricinus*, which is continuously expanding its latitudinal and altitudinal range in Europe [71]. Environmental factors play a vital role in the distribution of ticks, as most of *I. ricinus* lifecycle is spent off-host, where factors such as growth, reproduction, survival, and activity can be affected by environmental changes. It is predicted that the annual temperature of Europe will rise 1.5–2.5 °C in the coming few decades due to climate change, which may contribute to further expansion of tick distribution boundaries [72]. In more than 40% studies, ticks were collected from vegetation, so this habitat and the seasonal changes affecting it can be viewed as important. Free-living stages of *I. ricinus*, for example, require their vegetative habitats to retain 80% humidity to aid tick survival, with this, therefore, also promoting *B. burgdorferi* transmission. In contrast, areas with low humidity may reduce tick survival rates, activity, and distribution of *I. ricinus*. Understanding such microclimatic factors is crucial to understanding tick distribution and their role in the spread of the pathogens, and persistent monitoring is needed to observe the dynamic changes in tick habitats, the distribution of ticks, and the pathogens they carry.

Out of the 34 studies from Europe, 55% (*n* = 19) collected ticks from animals, with a high percentage of these (36.8%, *n* = 7) detecting *B. burgdorferi* in ticks from wild mammals (e.g., hedgehog, deer, brown bear, raccoon, and red foxes). Of the remaining studies, 21% (*n* = 4) collected *B. burgdorferi*-positive ticks from cats and dogs, 16% (*n* = 3) from birds, 10.5% (*n* = 2) from rodents, 10.5% (*n* = 2) from horses, and 5.2% (*n* = 1) from cows (Figure 3). Almost 95% (*n* = 18) of studies detected *B. burgdorferi* in *Ixodes ricinus*, which supports this tick’s importance as a major transmission risk of the pathogen in animals (Table 1). According to our literature review, a large proportion of studies reported that *B. burgdorferi* was circulating in ticks associated with wild animals, covering a vast range of hosts that could facilitate the movement of ticks. The contribution of wild animals in tick movement is also supported by another study conducted in the UK, where heavy infestation of ticks carrying *B. burgdorferi* were reported on gray squirrels [34]. The same authors also recovered ticks present on cats and dogs that were positive for this pathogen, thus posing a threat to the owners of these pets in terms of their risk of acquiring Lyme disease. According to a study by TickNET (a collaborative public health effort established by the CDC in 2007 which fosters coordinated surveillance, research, education, and prevention of tick-borne diseases), tick bite risk is increased nearly twofold through owning a pet [73], where companion animals that are allowed to roam freely can present a particular risk of bringing ticks into the home, creating both animal and public health concerns. Almost 16% of studies reported this pathogen’s detection in ticks from birds, among which *Ixodes scapularis* was the major vector after *Ixodes ricinus*. Infested birds, especially migratory birds, have potential to carry tick species over large distances, including from one continent to another, with this dispersal mechanism being at least partly responsible for increases in the distribution of ticks and the zoonotic pathogens they carry. According to a recent study, migratory birds were considered as a major factor in the expansion of *Ixodes scapularis* and its pathogen *Borrelia burgdorferi* [74]. As already discussed, climate change is another cause of enzootic transmission of *B. burgdorferi* and tick expansion, but yearly bidirectional migration of songbirds carrying ticks infected with zoonotic pathogens such as *B. burgdorferi*, *Borrelia mayonii, Borrelia miyamotoi, and Bartonella* in spring and fall may be even more significant, where it has been shown that birds infested with *Ixodes ricinus and Ixodes scapularis* can start new foci of this tick on islands [9,75,76,77,78] (Table 1).

#### 3.2.2. North America

In the previous two decades, a total of nine studies reported *B. burgdorferi* from ticks in North America; 88.9% (*n* = 8) studies reported this pathogen from the US and 11.1% (*n* = 1) from Canada. More than 50% (*n* = 5) of studies collected ticks from vegetation, and 44.4% (*n* = 4) from animals (e.g., pets, chipmunks, white-footed mice, dogs, and birds). Of these four, in one case, tick collection was from humans and found to be positive for *B. burgdorferi* (Figure 5 and Figure 6). PCR was used to confirm the presence of *Borrelia burgdorferi* in all of these studies and found an average of 19.2% prevalence of *Borrelia burgdorferi* in ticks of North America in all studies. In 66.7% (*n* = 6) of studies, the tick species testing positive for the pathogen was *Ixodes scapularis* (*I. scapularis)*, while in 22.2% (*n* = 2) of studies, it was *Ixodes pacificus* (*I. pacificus*). This demonstrates the significant contribution of both ticks for transmission of *B. burgdorferi* in North America (Table 1). According to a study from Michigan State University, *B*. *burgdorferi* was typically transmitted by black-legged ticks (*Ixodes scapularis*) in the east of the Rocky Mountains and by *I. pacificus* in the Western United States [79], which aligns with the findings of this review. A study in Canada based on passive surveillance data revealed that *I. scapularis* ticks are more common than previously suspected in this country [80]. In two studies from North America reviewed here, ticks collected from birds provided positive results for *B. burgdorferi*, with the role of migratory birds in spreading *B. burgdorferi* and *I. scapularis* reported in a study conducted in Ontario, Canada [81].

As with other tick species, climate change also exerts effects on the expansion of *I. scapularis* distributions; this is reported to be increasing where warmer conditions are prevailing [82], supporting claims of higher risks of Lyme disease in these areas in the future. Two studies have been conducted in North America which found *B. burgdorferi* in ticks collected from pets (cats and dogs), and in one study conducted in Ontario, Canada, pathogen-positive tick samples were collected from both humans and companion animals, reporting prevalence of the pathogen in *I. scapularis* at 17.5% and 9.9% respectively. The presence of ticks on companion animals is a significant risk factor for spread of the pathogens they carry. Studies considered in this review revealed the presence of *B. burgdorferi* carrying *I. scapularis* on pets, which is not only a source of transmission of *B. burgdorferi* to the pets themselves, but also poses a threat to humans with regard to Lyme disease transmission. The increasing population of black-legged ticks is also contributing to the transmission of other zoonotic pathogens such as *Anaplasma phagocytophilum, Babesia microti, Powassan virus*, and *Ehrlichia muris* [76]. In North America, this expansion of black-legged tick distribution, attributed to changes in land usage and climate change, is the major factor predicting the increased prevalence of zoonotic tick-borne diseases such as Lyme disease.

#### 3.2.3. Asia and Africa

In Asia, *B. burgdorferi* prevalence is quite low. In this review, China and Korea were the only countries in which this pathogen was reported in ticks, collected from vegetation and wild rodents, respectively. A study published in 2003 detected *B. burgdorferi* from Chinese *Ixodes persulcatus* collected from vegetation (Great Xingan Mountains, Small Xingan Mountains), with a prevalence of 33.8% (454/1345), while a 2020 study detected this pathogen in *Ixodes nipponensis, I. angustus*, and *H. longicornis* collected from wild rodents in Korea (Pocheon, Donghae, Sejong, Boryeong, Uiseong, Jeongup, Geoje, Goheung, and Jeju Island), with a prevalence of 33.6% (248/738) (Figure 5 and Figure 7) (Table 1).

*Ixodes persulcatus* are widely distributed from Russia to Eastern Asia, where one-fifth of the world’s human population resides. The study results conducted in China demonstrated that *B. burgdorferi* poses a health threat not only to animals, but also to humans, where *I. persulcatus* is prevalent [83]. *Ixodes persulcatus* is the most abundant tick species in China and is the major cause of tick bites in humans. *I. persulcatus* infests a range of nonhuman animal hosts as well, providing the opportunity to acquire more than one pathogen from different reservoirs [83]. In the case of Korea, the study reviewed here is the only one from this country evidencing *B. burgdorferi* in ticks, in this case taken from wild rodents, where these ticks had a high prevalence of pathogen, presenting a high risk of causing Lyme disease. Agriculture workers, hikers, and people living near tick-infested areas in Korea are thus at high risk of exposure to *Borrelia* due to proximity to wild rodents and the ticks they carry. Hence, continuous surveillance of tick species in various geographical regions of Korea can be considered important to minimizing possible disease transmission to humans.

In the case of Africa, prevalence of *B*. *burgdorferi* was unknown. Nevertheless, a recent 2021 study conducted in Egypt (Cairo, Giza, Al-Buhayrah, and Matrouh governorates) reported this pathogen in *Rhipicephalus sanguineus* ticks collected from dogs with a prevalence of 1.67% (Figure 6; Table 1). In Egypt, domestic animals are often highly infested with hard ticks. Although the rate of *B. burgdorferi* infection in dogs and ticks is low, dogs act as sentinel carriers for this pathogen. From a public health perspective, researchers should be aware of tick activity under various climatic conditions, which is often more than expected. The current data regarding *B. burgdorferi* transmission related to tick bites remains quite scarce, and its occurrence is thus likely underestimated.

### 3.3. Vector Control

Effective management of ticks is required for control of *B. burgdorferi*, as well as a range of other TBDs. *Ixodes ricinus*, for example, acts as a major vector responsible for spreading not only *B. burgdorferi*, but also other pathogens, including *Borrelia miyamotoi, Rickettsia slovaca, Rickettsia helvetica, Rickettsia monacensis, Anaplasma phagocytophilum, Babesia divergens, Babesia venatorum, Babesia microti, Bartonella henselae, Coxiella burnetii*, and *Francisella tularensis*, across the world [84]. In Europe, more than 90% of studies detected *B. burgdorferi* in this tick, which indicates the critical demand for control strategies against *I. ricinus*. Leveraging the low tolerance of this species for relative humidity levels below 85% could offer significant potential in managing this species and the diseases it spreads [85]. At low humidity, *I. ricinus* is unable to survive, and this intolerance can be used as a target to prevent tick infestations [86]. The second most important tick for *B. burgdorferi* transmission is *I. scapularis*, especially in the US. A retrospective review of *I. scapularis* has demonstrated significant range expansion over the last century in the US, which has had to be mitigated through appropriate control strategies to reduce the chances of transmission of *B. burgdorferi* [87].

There are various tick-control strategies used worldwide, many of which are associated with negative effects on the environment. A wide variety of chemical products in different compositions are effective against ticks such as *I. ricinus* and are commonly used to control ticks on domestic animals [88]. More environmentally considerate approaches are relatively rare, though advances in this space should be possible with increased research into delivering tick control through habitat management/manipulation, interrupting the tick lifecycle, or obtrusion of associated pathogenic transmission. In some cases, such measures should be relatively practical to deploy at scale, even utilizing existing animal management/husbandry techniques such as fencing, grazing, and mowing [89]. Nevertheless, the utility of environmental-friendly tick control approaches has received little attention, whilst, in contrast, the use of acaricides has been widespread. During the period from 1970 to 1980, for example, wide-ranging and extensive spraying of acaricides was carried out in Russia to control *I. persulcatus*, the main vector of the tick-borne encephalitis virus (TBEV) [90]. Such extensive acaricide use remains commonplace but is now increasingly considered as socially unacceptable, primarily due to the damaging effects of these chemicals on the environment and the biodiversity it contains [91]. Alternative and integrated approaches of controlling ticks should therefore be prioritized to reduce tick and TBD incidence on humans and animals whilst safeguarding the environment and better aligning to shifting societal needs. Design and development of such strategies is likely to benefit from cross-disciplinary collaboration, drawing from ecology, epidemiology, entomology, and the social sciences.

## 4. Conclusions

This review demonstrates that the number of tick species vectoring *B. burgdorferi* is increasing, reaching 19 to date. *Ixodes ricinus* is the most common tick found to be positive for this pathogen, in more than 70% of the studies considered, followed by *I. scapularis* (almost 19%). The wide distribution of these tick species is a concern, with this threatening to extend the geographic boundaries of emerging zoonotic diseases, including Lyme disease. Climate change and migratory birds with more exposure to ticks are playing a significant role in increasing the zoonotic transmission of *B. burgdorferi* across the world. Whilst recent research has clearly depicted this increased distribution (Figure 8), more comprehensive studies are still needed to better quantify the extent of this expansion and the prevalence of pathogens in tick species in some areas, especially on continents such as Asia and Africa. Advances in our understanding of effective nonchemical tick control measures are also needed if we are to address increasing threats from ticks and TBDs in an environmentally considerate manner.

## Figures and Tables

**Figure 1 microorganisms-09-02412-f001:**
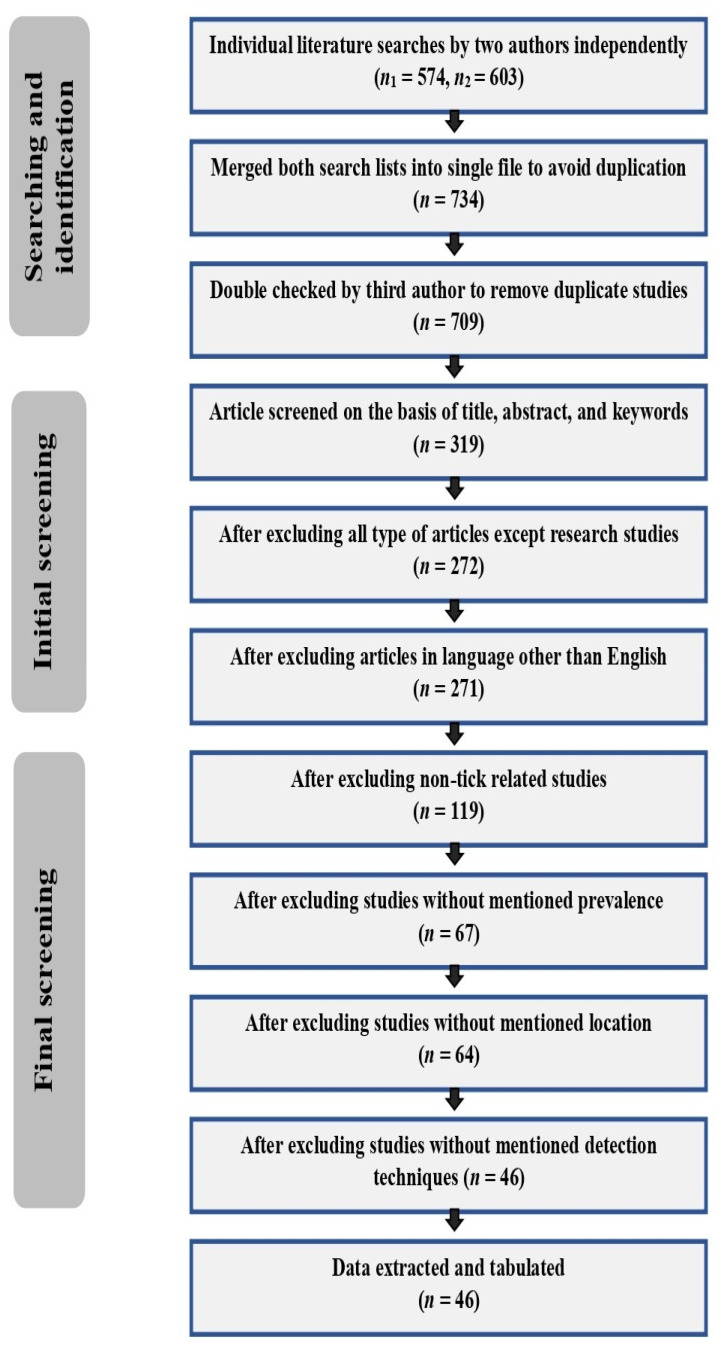
An overview of the selection procedure for studies recruited to this review according to PRISMA.

**Figure 2 microorganisms-09-02412-f002:**
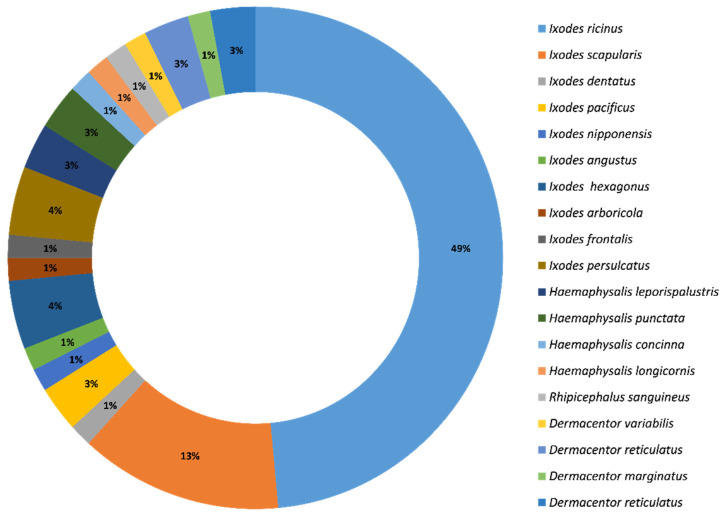
Reported proportions of tick species positive for *Borrelia burgdorferi* during the last two decades globally.

**Figure 3 microorganisms-09-02412-f003:**
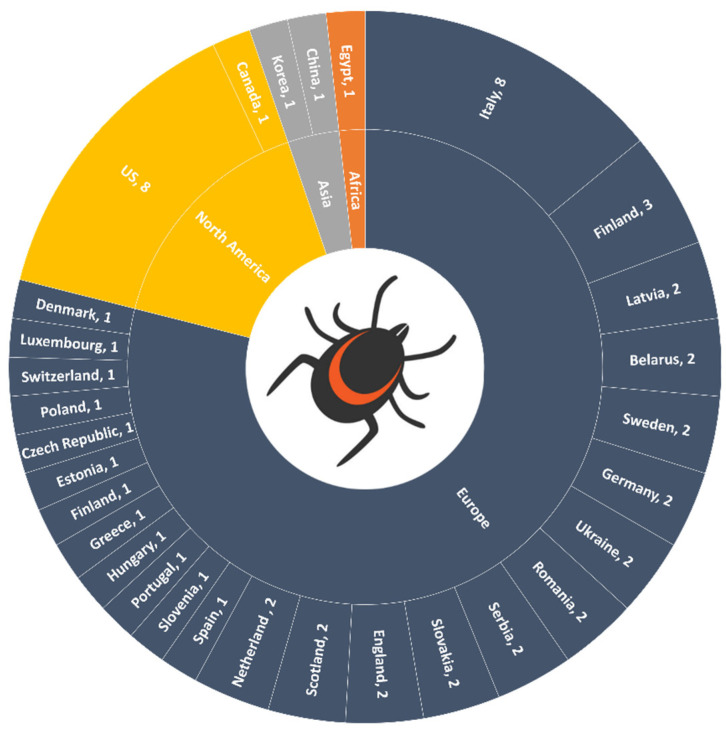
Number of studies reporting *Borrelia burgdorferi* in different tick species across the world during the last two decades.

**Figure 4 microorganisms-09-02412-f004:**
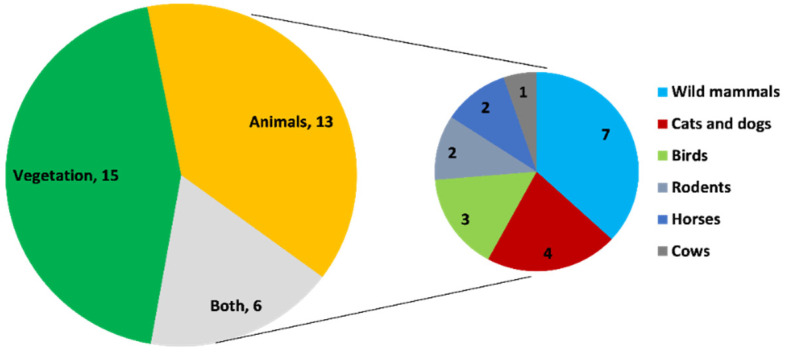
Number of studies on the basis of site of tick collection in Europe.

**Figure 5 microorganisms-09-02412-f005:**
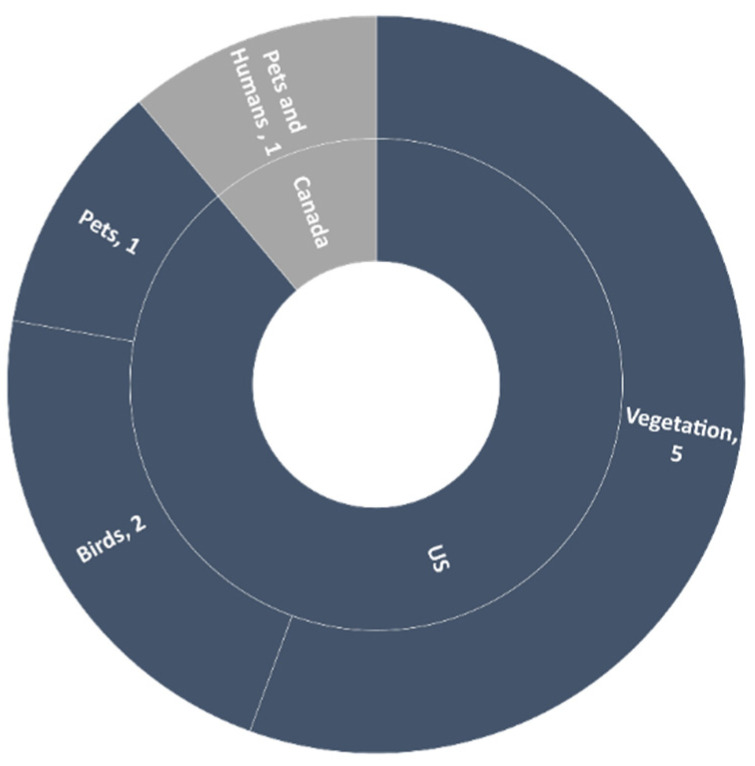
Number of studies on the basis of site of tick collection in North America.

**Figure 6 microorganisms-09-02412-f006:**
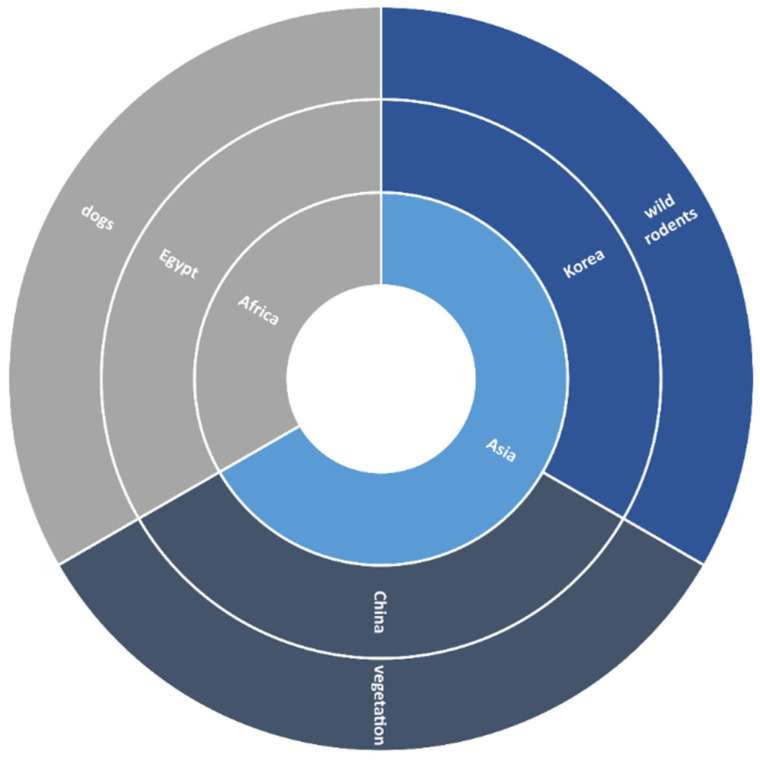
Number of studies on the basis of site of tick collection in Asia and Africa.

**Figure 7 microorganisms-09-02412-f007:**
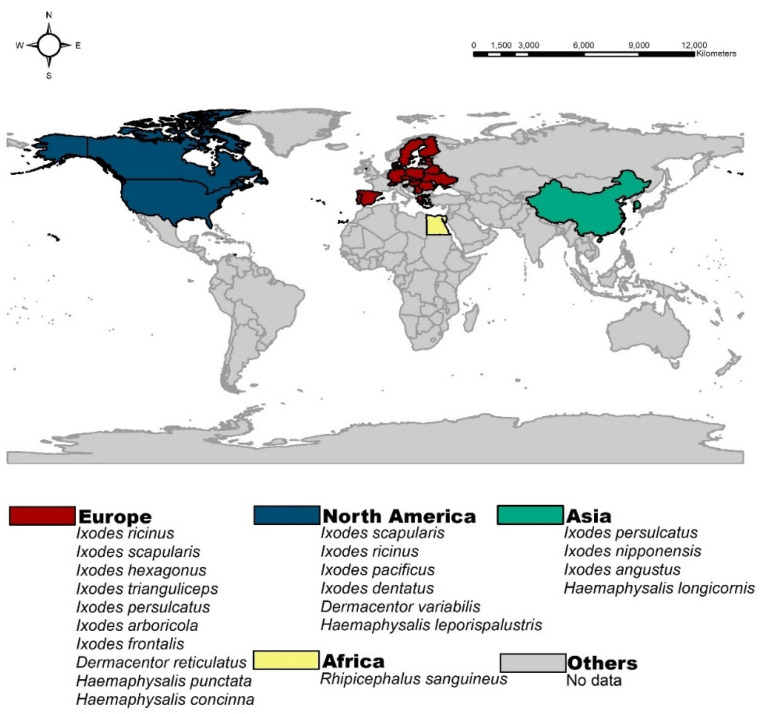
Geographical distribution of tick species carrying *Borrelia burgdorferi* across the world.

**Figure 8 microorganisms-09-02412-f008:**
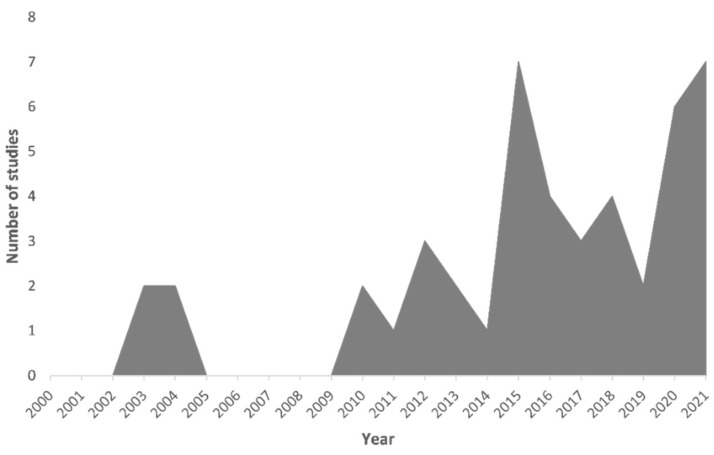
Timewise studies focusing on tick for *Borrelia burgdorferi* in the last two decades.

**Table 1 microorganisms-09-02412-t001:** Literature focusing on detection of *Borrelia burgdorferi* in ticks during the last two decades across the world.

Continent	Country	Region	Host/Sampling Site	Tick Species	Total Ticks Collected	Positive (*n*)	Prevalence %	95% CI	Molecular Technique	Year of Study	Refs.
**Europe**											
Europe	Slovakia	Western Slovakia	Vegetation (Parks)	*Ixodes scapularis*	1294	420	33%	Not given	PCR	1999–2000	[24]
			Birds (*Parus major*, *Turdus merula*, *Turdus philomelos*)	*Ixodes scapularis*	57	16	28%	Not given	PCR		
	Serbia	Vojvodina	Vegetation (Forest)	*Ixodes ricinus*	764	169	22.1%	11–29	PCR	2006–2008	[25]
	Luxemburg	Not mentioned	Vegetation (Forest)	*Ixodes ricinus*	1394	157	11.3%	Not given	PCR	2007	[26]
	Switzerland	11 sites located between 400 and 900 m above sea level	Vegetation (Parks)	*Ixodes ricinus*	1458	328	22.5%	Not given	PCR	2009–2010	[27]
	Sweden	Southern Sweden	Rodents (*Myodes glareolus, Apodemus flavicollis*)	*Ixodes ricinus*	276	137	49.6%	Not given	q-PCR	2008-2010	[28]
	Belarus	Brest, Gomel, Grodno, Minsk, Mogilev, Vitebsk	Vegetation (Parks)	*Ixodes ricinus*	553	52	9.4%	Not given	PCR	2009	[29]
	Italy	Borzonasca, Chiavari	Vegetation (Forest)	*Ixodes ricinus*	170	31	18.2%	Not given	PCR	1998–1999	[30]
	Denmark	South Jut land	Dogs	*Ixodes ricinus*	661	99	15%	Not given	PCR	2011	[31]
	Italy	Emilia-Romagna	Vegetation (Forest)	*Ixodes ricinus*	284	78	27.5%	Not given	Real-time PCR	2010	[32]
	Italy	Ossola Valley Province of Verbano–Cusio–Ossola	Vegetation and Wild animals (chamois, roe deer, red deer)	*Ixodes ricinus*	1766	530	30%	Not given	PCR	2011	[33]
	Scotland and Northern England	Not mentioned	Gray squirrel	*Ixodes ricinus*	1585	189	11.9%	9.7–14.6	PCR	2012–2013	[34]
	Italy	Belluno, Perugia	Vegetation (Forest)	*Ixodes ricinus*	447	17	3.8%	Not given	PCR	2007–2010	[35]
	Italy	Tuscany	Wild animals (*Dama dama, Cervus elaphus*)	*Ixodes ricinus*	420	6	1.4%	Not given	PCR	2015–2019	[36]
	Netherlands	Flevoland, Gelderland, Noord-Holland, Utrecht, and Zuid-Holland	Hedgehogs	*Ixodes ricinus*	460	67	14%	Not given	q-PCR	2010–2014	[37]
	Netherlands	Not mentioned	Horse	*Ixodes ricinus*	120	52	43.3%	Not given	PCR	2018	[38]
	Serbia	Forests (Lipovica, Bojčinska, Avala, Miljakovačk, Makiš), Park-forests (Ada Ciganlija, Zvezdara, Banjica, Košutnjak, Jajinci) Parks (Hajd park, Belevode, Usće, Šumice, Kalemegdan, Topčider, Tašmajdan, Banovobrdo, Pionirski park)	Vegetation (Forest)	*Ixodes ricinus*	3199	704	22%	Not given	PCR	2009	[39]
	Finland	Southwestern Finland	Vegetation (Forest)	*Ixodes ricinus*	3169	217	6.8%	Not given	PCR	2013–2014	[40]
	Poland	Goleniowska Forest	Shetland ponies	*Ixodes ricinus*	1737	333	19%	Not given	PCR	2010–2012	[41]
			Vegetation (Parks)	*Ixodes ricinus*	371	18	4.8%	Not given	PCR		
	Romania	Eastern Romania	Vegetation (Forest)	*Ixodes ricinus*	534	138	25.8%	Not given	PCR	2014	[42]
	UK	Not mentioned	Cat	*Ixodes ricinus, Ixodes hexagonus, Ixodes trianguliceps*	541	15	2.8%	Not given	PCR	2016	[43]
	Germany	Saxony	Small mammals (*Apodemus agrarius, Apodemus flavicollis, Microtus arvalis, Microtus agrestis, Mustela nivalis, Myodes glareolus Sorex araneus*, *Talpa europaea*)	*Ixodes ricinus*	2802	154	5.5	3.5–8.3	PCR	2012–2016	[44]
	Slovakia	Bratislava	Birds (*Parus major*, *Sitta europaea*, *Turdus merula*, *Erithacus rubecula*, *Dendrocopos major*, *Parus montanus*, *Fringilla coelebs*, *Parus caeruleus*, *Muscicapa striata*)	*Ixodes ricinus*	295	37	12.5%	Not given	PCR	2011–2012	[45]
	Italy	Dolomiti Bellunesi National Park in the Province of Bellun	Red foxes (Parks)	*Ixodes ricinus*	2248	28	1.25%	Not given	Real-time PCR	2011–2016	[46]
	Scotland	Loch Lomond and Trossachs National Park	Vegetation (Forest)	*Ixodes ricinus*	6567	91	1.4%	1.1–1.7	PCR	2011–2015	[47]
	Latvia	Not mentioned	Dog	*Ixodes ricinus*, *Dermacentor reticulatus*	608	48	7.9%	Not given	Nested-PCR	2011–2016	[48]
	Italy	Aosta Valley, western Alps	Vegetation (Forest)	*Ixodes scapularis*	30	12	40%	22.5–57.5	PCR	2016	[49]
	Latvia	All regions of Latvia	Vegetation (Parks)	*Ixodes ricinus, Ixodes persulcatus, Dermacentor reticulatus*	4593	657	14%	Not given	PCR	2017–2019	[50]
	Italy	64 Italian provinces	Dog	*Ixodes ricinus, Ixodes hexagonous*	723	3	0.4%	0.2–0.8	PCR	2016–2017	[51]
	Finland	8 sites on the coast of Bothnian Bay	Vegetation (Forest)	*Ixodes persulcatus*	163	101	62%	55–70	PCR	2019	[52]
	Czech Republic, Estonia, Finland, Germany, Greece, Hungary, Netherlands, Portugal, Slovenia, Spain and Sweden	11 European countries	Birds	*Ixodes ricinus, Ixodes arboricola, Ixodes frontalis*	656	244	37.2%	Not given	PCR	2005–2008 2013–2014 and 2016	[53]
	Ukraine	Chernivtsi, Khmelnytskyi, Kyiv, Ternopil, Vinnytsia regions	Vegetation, wild and domestic animals (brown bear, raccoon, red fox, lynx, cats, cattle dogs)	*Ixodes ricinus*	99	25	25%	Not given	PCR	2019–2020	[54]
	Belarus	Brest Gomel, Grodno, Minsk, Mogilev and Vitebsk	Vegetation and Cow	*Ixodes ricinus, Dermacentor reticulatus*	4070	253	6.2%	Not given	PCR	2012–2019	[55]
	Romania	Luliu Haţieganu Park, Alexandru Borza Botanical Garden, Mănăştur Cemetery Hoia, Făget forest	Vegetation (Forest)	*Ixodes ricinus, Haemaphysalis punctata*	148	39	26.35%	19.46–34.22	PCR	2018	[56]
			Rodents, birds, and hedgehogs	*Ixodes ricinus*, *Ixodes hexagonus, Haemaphysalis punctata, Haemaphysalis concinna*	222	81	36.5%	29.29–42.27	PCR		
	Ukraine	Southeastern Ukraine (Zaporizhzhya region)	Vegetation (Forest)	*Ixodes ricinus*	358	115.6	32.3%	Not given	PCR	2014–2018	[57]
**North America**											
North America	US	Southern coastal Maine	Pets, chipmunks, white-footed mice	*Ixodes scapularis*	394	88	22.3%	Not given	PCR	1995–1997	[58]
	US	University of California Hopland Research and Extension Center (HREC)	Vegetation (Forest)	*Ixodes pacificus*	181	7	3.9%	Not given	PCR	2003	[59]
	US	Southwestern Michigan	Birds	*Ixodes dentatus, Haemaphysalis leporispalustris, Ixodes scapularis, Dermacentor variabilis*	12,301	517	4.2%	Not given	PCR	2004–2007	[60]
	US	Southwestern suburban Chicago	Wild birds	*Ixodes scapularis, Haemaphysalis leporispalustris*	120	5	4%	Not given	PCR	2005–2010	[61]
	US	Hudson Valley	Vegetation (Forest)	*Ixodes ricinus*	1245	760	61%	Not given	PCR	2011	[62]
	US	New Castle County, Delaware	Vegetation (Parks)	*Ixodes scapularis*	441	46	10.4%	Not given	PCR	2013–2014	[63]
	US	New York State	Vegetation (Forest)	*Ixodes* *scapularis*	677	346	51%	39.3–63.3	rt-PCR	2018	[64]
	US	Marin County California	Vegetation (Parks)	*Ixodes pacificus*	1419	41	2.9%	2.3–3.7	rt-PCR	2015–2018	[65]
	Canada	Ontario	Human	*Ixodes* *scapularis*	17,230	3015	17.5%	16.97–18.09	PCR	2011–2017	[66]
			Companion animals (dogs)	*Ixodes scapularis*	4375	433	9.9%	9.15–10.78	PCR		
**Asia**											
Asia	Korea	Pocheon, Donghae, Sejong, Boryeong, Uiseong, Jeongup, Geoje, Goheung, and Jeju Island	Wild rodents	*Ixodes nipponensis*, *Ixodes angustus*, *Haemaphysalis longicornis*	738	248	33.6%	Not given	PCR	2017	[67]
	China	Great Xingan Mountains, Small Xingan Mountains	Vegetation (parks)	*Ixodes persulcatus*	1345	454	33.8%	Not given	PCR	1999–2001	[68]
**Africa**											
Africa	Egypt	Cairo, Giza, Al-Buhayrah, and Matrouh govern	Dog	*Rhipicephalus sanguineus*	60	1	1.67%	Not given	PCR	2017	[69]

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
