# Peer review of "The Role of Ticks in the Emergence of Borrelia burgdorferi as a Zoonotic Pathogen and Its Vector Control: A Global Systemic Review"

_microorganisms, 2021, doi:10.3390/microorganisms9122412_

Round 1
Reviewer 1 Report
Article is interesting and good written. Authors present systematic review of 46 studies, in which search for species of ticks responsible for Borrelia burgdorferi transmission worldwide. Some suggestions:
- Figures are too small and very often descriptions are illegible.
- For the readers can be important if vegetations were lawn, meadow, low shrubs, deciduous forest, mixed forest, or coniferous forest. Please add this information from cited literature.
Author Response
Dear Reviewer,
Enclosed here attachment that contains all answers
Thank you so much for your valuable additions and kind suggestions

Reviewer 2 Report
I have read with interest this systematic review where authors evaluated and synthetized data on which tick species, and in which continents are responsible for transmission of B. burgdorferi. This study is comprehensive, well written and I believe it will contribute to existing knowledge on this topic. I do have the following major comments that I would like to ask author to address:
- The aim of the study was not listed in the abstract.
- In introduction- it should be mentioned that due to climate change the incidence of Lyme disease in USA is expected to increase for about 20% in the next 1-2 decades ( https://pubmed.ncbi.nlm.nih.gov/30473737/
- Introduction when you mention B afzelii and B. garinii species- it should be stated that these pathogens are common in Europe ( unlike in USA), and authors should clearly mention this. Furthermore, these species have been implicated to cause more distinct clinical syndromes like Bannwarth, which we rarely see in USA. On the other hand, arthritis as a late Lyme complication , while very common in USA is rare in Europe- which is of interest ( https://www.ncbi.nlm.nih.gov/pmc/articles/PMC6317275/
- Methodology- I suppose that authors reviewed references of the articles they finally selected. This was not mentioned in methodology and as it is a part of PRISMA this should be clarified.
- Results section 3.2.2- North America- ref 30 is obsolete, from Russia and should be removed. As I suggested above, newer reference from USA from 2018 is available and should be cited
- In the same section 3.2.2 authors incorrectly mention the other pathogens that can be transmitted through Ixodes tick. In addition to the ones they mention, Borrelia mayonii, Borrelia miyamotoi and Bartonella should be added ( https://www.ncbi.nlm.nih.gov/pmc/articles/PMC5879012/, https://pubmed.ncbi.nlm.nih.gov/28285589/, https://www.ncbi.nlm.nih.gov/pmc/articles/PMC2600320/
Author Response
Dear Reviewer and editor
Enclosed here attachment, all your kind suggestions and comments have been addressed and cited in the manuscript as per your suggestions
Thank you so much for your valuable additions

Round 2
Reviewer 2 Report
The authors have responded to all of my questions successfully.